# From Oncological Paradigms to Non-Communicable Disease Pandemic. The Need of Recovery Human Biology Evolution

**DOI:** 10.3390/ijerph181910087

**Published:** 2021-09-25

**Authors:** Elia Biganzoli, Romano Demicheli

**Affiliations:** Unit of Medical Statistics, Biometry and Epidemiology, Department of Biomedical and Clinical Sciences (DIBIC) “L. Sacco” & DSRC, LITA Vialba Campus, Università degli Studi di Milano, Via Gian Battista Grassi, 74-20157 Milano, Italy

**Keywords:** cancer dormancy, low-grade chronic inflammation, healthy aging, physical exercise

## Abstract

The paradigm of the Somatic Mutation Theory (SMT) is failing, and a new paradigm is underway but not yet established. What is being challenged is a conceptual approach that involves the entire human biology and the development of chronic diseases. The behavior of breast and other solid cancers is compatible with the concept that the primary tumor is able to control its microscopic metastases, in the same way that an organ (e.g., the liver) is able to control its physiological size. This finding suggested that cancer and its metastases may behave as an organoid. The new paradigm under construction considers the origin of tumors as a disturbance in the communication network between tissue cell populations and between cells and extracellular matrix, and supports a systemic approach to the study of both healthy and pathologic tissues. The commentary provides a rationale for the role of physical exercise in the control of tumor dormancy according to a human evolutionary perspective.

## 1. Introduction

Paradigms are conceptual tools, i.e., reasoned descriptions of what we believe reality should be. Science history is a chain of paradigms guiding behaviours and research, which are able to provide a temporary useful picture of reality but are usually destined to face failures [1], forcing scientists to devise new paradigms, which are valuable until they fail in turn. It is very interesting and stimulating for a researcher acting during the time of transition from an old failing paradigm to a new, emerging and not yet well-defined one. This window of uncertainty encourages the exploration of new viewpoints. As cancer biologists and oncologists, the writers are experiencing one of these transitional moments: the old paradigm, called Somatic Mutation Theory (SMT), is failing and a new paradigm is in the making but not yet established. This process is not limited to oncology, because what is being challenged in this specific field is a conceptual approach that involves the entire human biology and the development of chronic diseases.

## 2. The Somatic Mutation Theory (SMT) in Cancer

The central idea of SMT, which was proposed by Theodor Boveri in 1914 [2], posits that the root cause of cancer is a mutation of the genetic material of a normal cell, an event producing a “transformed” cell capable of uncontrolled proliferation. This idea gained impetus from the molecular biology revolution and became the generalized approach to cancer research. The SMT has been fully described (and supported) by Hanahan and Weinberg [3], who also suggested justifications for some discrepancies between theory and the experimental data. In short, it is accepted that cancer is a pathological phenomenon occurring g at the cellular level, which originates in the genome through a sufficient number of mutations, amplifications and/or deletions in crucial genes. In the original paradigm, the gene alterations make the cell independent of the external context, which is a “helpless bystander”, forced to provide support for tumour progression. Cell transformation is an irreversible process: “Once a cancer cell, always a cancer cell”, implying that the only way to control the disease is the eradication of all neoplastic cells. The SMT has driven oncology research for decades, in tune with the idea that all biological behaviours play out at the gene level, an idea that abundantly pervaded large areas of biological research. However, despite the substantial public and private funding (mostly allocated to cancer research) and the enormous technological developments in biological research, by the end of the century, the “fighting” promises of the “War on Cancer” had been largely betrayed, as even its supporters admit [4]. In addition, and more importantly, it should be considered that the number of failures of this gene-centric paradigm in explaining biological phenomena and experimental data has become cumbersome. No mutations common to the various tumour types are yet known, despite the enormous investigation effort in this regard, leaving the question open [5]. Tumours may occur in the absence of gene alterations [6,7]. The formation of carcinomas can occur by carcinogen treatment or by irradiation of the stroma alone [8,9]. The asserted irreversibility of neoplasms also contrasts with established clinical observations, such as the spontaneous regression of neuroblastomas, including metastatic ones [10]. In addition, there are numerous experimental data in animals documenting phenotypic normalization of neoplastic cells despite their genomic configuration remaining unchanged [7,11,12,13,14]. Furthermore, SMT essentially ignores the host, who is another player in the clinical history of people suffering from cancer, and capable of establishing a balance with the disease in its etiopathogenesis and clinical history, such that it allows some mammary carcinomas to remain dormant for life while others grow rapidly to kill the host [15,16].

## 3. Cancer Dormancy Leads to a New Paradigm

Since the beginning of the SMT domain, critical voices have suggested that the “bad-cell”-centric explanation of cancer is unable to describe the essence of the tissue structure, which is based on the relationship between cells, and proposed other explanations focused at the tissue level [17,18,19,20,21]. The Smithers’ metaphor is particularly iconic: “A lifetime of studying the internal combustion engine would not help anyone understand our traffic problems”. We ourselves shared the need to address the behaviour of cancer at the tissue level when we were faced. at the clinical level, with the phenomenon of tumour dormancy [21,22], which, until then, was confined to specific areas of experimental research [23]. Indeed, the behaviour of breast cancer is very compatible with the concept that the primary tumour is able to control its microscopic metastases, in the same way that an organ (e.g., the liver) is able to control its physiological size. This finding suggested that breast cancer and its metastases may behave as an organoid [24,25]. Just as a partial hepatectomy induces the proliferation of hitherto “dormant” liver cells, surgical removal of the primary induces the emergence of its subclinical metastases from the dormancy in which they were maintained, according to the cancer homeostasis. Moreover, the dynamics of metastasis emergence at the clinical level reveals that dormancy states may have different biological characteristics, conditioning the timing of emergence during follow-up [26,27]. In summary, the new paradigm under construction considers the origin of tumours as a disturbance in the communication network between tissue cell populations and between cells and the extracellular matrix, and supports a systemic approach to the study of both healthy and pathologic tissues, according to adaptive cell and tissue physio-metabolic perspectives. The questions raised involve the causes of this disorder, how it can be managed and how prevention and treatment be individualized according to the new paradigm.

Since normal tissues are the end result of a process of self-organization [28], where architectures emerge from cell populations originally lacking a particular order [29], there is no main reason why we should not accept that similar mechanisms are, at least in part, conserved for cancers. In particular, according to this systemic approach, processes are characterized by nonlinearity, multilevel interactions, and the dependence of each architectural change on its previous history [30,31]. This is clearly an alternative view to the dominant one in molecular biology, where it is assumed that the phenotype of a cell is caused by linear chains of events along defined molecular lines (pathways) of interaction. This approach looks almost indispensable if we consider the metastasis-host system, in which the evolution can range, as mentioned above, from rapid and lethal emergence of disease recurrence to very long periods of disease-free survival, where dormancy is possibly prolonged for the entire lifetime [14].

## 4. Physical Exercise and Cancer Dormancy

In particular, the need for a systemic approach further emerged from the observation that a physiological/metabolic activity realized by targeted physical exercise interventions may improve the disease outcomes in certain subsets of patients [32,33]. Of note, according to a recent analysis of the disease dynamics, this improvement emerges after about 5 years after primary treatment [34], suggesting that physical activity may exert its long-term effects on given types of dormant metastases during their subclinical development. These findings are in agreement with the hypothesis that skeletal muscle behaves as an immune/endocrine organ, capable of secreting cytokines into the circulatory system during physical activity, a finding that could even associate active skeletal muscles with the maintenance of a healthy immune system during ageing [35]. Of note, a reduction in levels of inflammatory markers is observed during physical activity [36,37]. Therefore, physical exercise might act on immune-related dormancy through its modulatory effects on the immune processes, possibly related to chronic inflammation. In turn, immune processes are related to the balance in the optimal management of energy resources across different tissues and organs at the individual level, borrowing social reflections according to the evolutionary perspective of homeostasis. In any case, this is a system-wide phenomenon that again underscores the need for a non-reductionist approach to many problems, and to frame them within a broader horizon.

Cancer is a non-communicable disease and follows the dynamic of other chronic-degenerative diseases, in terms of both its frequency and impact on life. The incidence of various diseases and the human life expectancy revealed a drastic change during the 20th century, particularly its last decades [38]. Interventions on communicable diseases (antibiotics, vaccines, improved hygiene, better lifestyle habits) have substantially reduced mortality in the early and middle stages of life in high-income countries and have the potential to achieve similar results in low- and middle-income countries. However, a change in the most widespread pathologies has been concomitantly observed, with a significant increase in chronic-degenerative diseases such as cancer, heart disease, neurodegenerative conditions, osteoporosis, arthritis, diabetes, sarcopenia, which are prevalent in high-income countries but also spreading in others. Contrary to expectations, however, an increase in life expectancy was observed at older ages (>80 years) [39], stressing that we are faced with a complex and multifactorial phenomenon, for which a series of possible explanations with different theoretical and experimental basis can be advanced.

## 5. Human Evolution and Chronic Diseases

From an evolutionary perspective, the physiological structure that distinguishes contemporary humans is the result of a process of optimization of the relationship between “nature and nurture” or, in more stringent terms, between “genome and environment”. If we retrace the history of Homo sapiens and his ancestors, we realize that, after a time period of the order of millions of years, the conditions of life have greatly changed since the Agricultural Revolution (about ten thousand years) and, in a much more dramatic way, in the last two centuries, with the Industrial Revolution [40]. Thus, the equilibrium achieved during essentially constant conditions, at least during the Palaeolithic, was broken in a very short time compared to the full biological evolutionary perspective. This is due to the sudden changes in diet, physical activity, stress, environmental pollution, etc., that led to the disruption of homeostatic balances acquired in previous millennia, which is the origin of chronic degenerative diseases [41]. This conflict between our Palaeolithic biology and the environmental conditions of consumer societies is thought to have led to, as a main consequence, the chronic low-grade inflammation and metabolic syndromes that are correlated with many of the non-communicable diseases we are discussing [42], as well as other major global sustainability issues. 

One could wonder if the expansion of chronic degenerative diseases that characterizes our times corresponds to an evolutionary process already in place in the Paleolithic era. If we consider such pathologies in a society where mortality was dominated by accidental causes or transmissible diseases, we could hypothesize that they worked as a selection for individuals capable of healthy aging. The healthy aging people were functional to support the group to which they belonged, partly with their preserved activity as hunter-gatherers and partly with their activity as a depository of experience and cultural background. Any drift from the optimized physiometabolic balance of the hunter–gatherers was possibly damaging, even for the survival of the social group, according to the principle that unfitted surviving individuals were wasting resources outside their functional biological expectation, leading to the homeostatic switch to low-grade chronic inflammation with accelerating aging and death. Hence, physical activity could be considered a component and biomarker of the homeostatic systems that ensure the maintenance of the state of health, functional both to individual life and to the fitness of the group in hunter–gatherer biology, which is still largely preserved. Its significant reduction in current societies, therefore, has impacts on both levels, and its highlighted effect on breast cancer prognosis [32,33,34] could be interpreted as a partial recovery of this homeostatic function linked to the deep biological need for healthy aging, with its social reflection.

## 6. Sustainability Perspectives

Much research has been conducted and is in progress on the phenomenon of change in the pathological landscape that is configurable as a true “non-communicable epidemic” [42]. In particular, for individual diseases, risk factors have been identified that show extensive overlap with each other. Particular attention has been given to diet, for which conditions of high risk have been identified in the use of foods with high protein and lipid content, especially industrial ultra-processed food [43]. An examination of this topic is beyond the scope of this commentary. We want to emphasize here, however, that the problem cannot be addressed in a reductionist way, since it involves many interactions between different factors. An example of this can be found in the studies on the Masai of East Africa, which show that, despite a diet rich in proteins and fats and poor in carbohydrates [44], cardiovascular diseases are almost absent [45], a fact that can be linked to the physical fitness linked to their way of life [46]. The very high observed interconnection between the different levels of influence of diet, exercise, lifestyle, etc., supports the need for a systemic approach to investigations. Integrating the different factors involved is certainly difficult, but inevitable, as is a similar therapeutic approach to such diseases. Single and sectoral measures (less sugar in drinks, reduction in smoking, fewer hours at the computer, etc.) are fundamental but still modest palliatives compared to the global dimensions of the problem. The main road to reach significant health goals is still a comprehensive and coordinated sustainability change in economic organization and lifestyles, which, at present, mainly function for the globalized consumer society. The recovery of physical exercise, from an anthropological perspective, according to the long-standing evolution of human biology is an urgent necessity. 

## Data Availability

Not applicable.

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
