# Peer review of "From Oncological Paradigms to Non-Communicable Disease Pandemic. The Need of Recovery Human Biology Evolution"

_ijerph, 2021, doi:10.3390/ijerph181910087_

Round 1

Reviewer 1 Report

This like commentary show a new scenario about the role of physical exercise in the control of tumor dormancy according to a human evolutionary perspective. The authors was able to clarify that within the disease there are several instrinsic and estrinsic factors while is needed the phyisical exercise

Author Response

We thank the anonymous Reviewer for the positive comments.

Reviewer 2 Report

The authors discuss an important and interesting aspect of oncology. Despite the existence of paradigms, scientific knowledge has always allowed its evolution. The hallmarks punctuated by Hanahan are interconnected in a complex network of events that contribute to the development and progression of tumors. In this document, despite the authors recognizing the importance of the microenvironment in the development of tumors, some aspects were not properly explored. The authors pointed that “In summary, the new paradigm under construction considers the origin of tumors as a disturbance in the communication network between tissue cell populations and between cells and extracellular matrix, and supports a systemic approach to the study of both healthy and pathologic tissues .” What would cause this disorder? How can this disorder be treated? How can treatment be individualized?

The authors bring, then, the importance of physical exercises in the modulation of this communication network, however, they do not discuss the subject from a new perspective, in addition to the one already published. Therefore, I recommend adding new points discussing the importance of exercises from a new perspective.

Author Response

We thank the anonymous Reviewer for the relevant comments pointing out major horizons of current system research in cancer and chronic diseases.

We shared the questions related to the origin of tumors as a disturbance in the communication network between tissue cell populations and between cells and extracellular matrix, supporting a systemic approach to the study of both healthy and pathologic tissues, namely: what would cause this disorder? How can this disorder be treated? How can treatment be individualized?

However, answering to these question will be the actual challenge of future research on this topic. Therefore, we embed this theme in the text including the following statement:

"In summary, the new paradigm under construction considers the origin of tumours as a disturbance in the communication network between tissue cell populations and between cells and extracellular matrix, and supports a systemic approach to the study of both healthy and pathologic tissues, according to adaptive cell and tissue physio-metabolic perspectives. The rising questions involves the causes of this disorder, how it can be managed and how prevention and treatment be individualized according to the new paradigm."

Concerning the request of new points discussing the importance of exercises from a new perspective, we added the following consideration:

"In turn, immune processes are related to the balance in the optimal management of energy resources across different tissues and organs at the individual level, borrowing social reflections according to the evolutionary perspective of homeostasis."

A more extensive review would be outside the scope of the commentary.

Reviewer 3 Report

The authors of this commentary pretend to generate a vision of paradigms of diseases, the difficulties of its evolutions, different influences that make its study and prevention, a challenge. The commentary in particular focuses on the cancer paradigm, explaining the Somatic Mutation Theory and other aspects of this disease in order to understand the evolution and war on cancer.

Generally speaking, the comment is quite appropriate, however, some aspects can be improved:

- In the abstract, the authors seem that they focus on breast cancer in particular when the comment later is not so specific.

- In relation with the with the distribution of the sections the section number 5 “Human evolution and chronic diseases”, I think it would be more logical at the beginning of the comment, since it is more general.

- The incorporation of some figures, tables or schemes would make reading this comment more striking.

Author Response

We thank the anonymous Reviewer for the positive comments and suggestions.

  • In the abstract, the authors seem that they focus on breast cancer in particular when the comment later is not so specific.
    • R: In the abstract we refer now to solid cancers although the main cited literature examples are related to breast cancer
  • In relation with the with the distribution of the sections the section number 5 “Human evolution and chronic diseases”, I think it would be more logical at the beginning of the comment, since it is more general.
    • R: we would like to keep the present organization of the sections according to the generalization of the aspects related to cancer. The additional statements we provided should support this choice
  • The incorporation of some figures, tables or schemes would make reading this comment more striking.
    • R: we were not forecasting this option which would be certainly appealing since the commentary nature of the paper. We could possibly adapt figures or schemes from our previous papers but it could involve publication rights.